# Are Synthetic Time-series Data Really not as Good as Real Data?

## Abstract

To alleviate the commonly encountered inadequate time-series data problem in DL (DL), we develop a non-DL generic data synthesis method. When current methods require real data or data statistics to train generators or synthesize data, our method InfoBoost enables zero-shot training of models without the need for real data or data statistics. Additionally, as an application of our synthetic data, we train an unconditional feature (rhythm, noise, trend) decomposer based on our synthetic data, which is applicable to real time-series data. Through experiments, our non-DL synthetic data enables models to achieve superior performance on unsupervised tasks and self-supervised prediction & imputation compared models using real data. Visualized case studies further demonstrate the effectiveness of our novel unconditional feature decomposer trained with our synthetic data.

## 1 Introduction

DL (DL) practitioners across various time-series domains commonly encounter the obstacle of inadequate data, domains including finance Tang et al. (2022), energy P et al. (2022), traffic Shaygan et al. (2022), weather Zhu et al. (2023), and healthcare Saeidi et al. (2021); Wang et al. (2022); Zhang et al. (2023). Procuring sufficient time-series data is often financially costly, which requires substantial human effort for data preprocessing, and entails privacy concerns involving commercial or personal information.

In response to this concern, we introduce InfoBoost, a framework designed to produce general-purpose synthetic time-series data. By harnessing the capabilities of non-DL synthetic data generation, InfoBoost offers a universally applicable solution that bypasses the expenses and privacy risks associated with obtaining and handling real-world time-series data for model training.

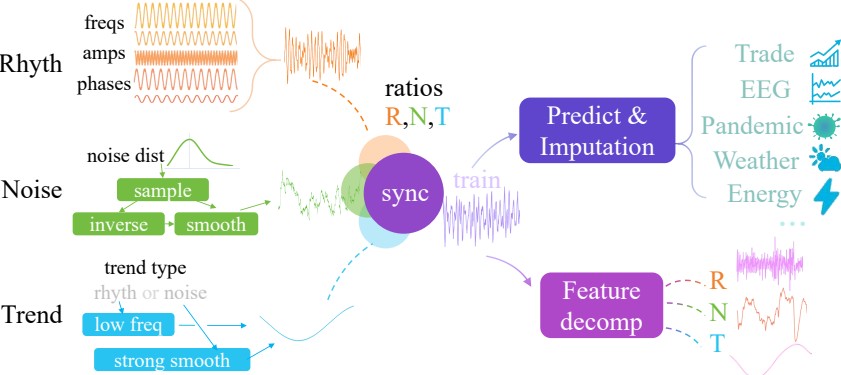

Figure 1: Schematic of the InfoBoost model illustrating the data synthesis, feature decomposing learning, and feature extraction processes. In the diagram, the label 'Rhyth' corresponds to the multi-source rhythmic data (MRD), 'Noise' corresponds to different types of noise and their noise ratios (TN & NR), and 'Trend' corresponds to trend information (TI). The visual features elucidate the individual roles of each component within the InfoBoost framework. 'Sync' stands for 'synthesized data', representing data that are artificially generated, integrating MRD, TN & NR, and TI.

Inspired by the potential benefits of synthetic data Savage (2023); Luo et al. (2023); Yin et al. (2023b); Zhang et al. (2024), we explored synthetic time-series data generation methods. However, we observed that almost all time-series data generation approaches require sampling training data from real datasets or fine-tuning DL-based generators Luo et al. (2023); Yang et al. (2022); Yang & Hong (2022). While existing methods can somewhat alleviate data quality, bias, and vulnerability issues, they still require ample amounts of high-quality real data from the specific domain to ensure the generator's output encompasses unseen data, thereby unable to fully overcome the challenge of inadequate if certain time-series domain real data. Therefore, we believe that to address the limitations of DL-based data synthesis, a more generic time-series data synthesis method that does not rely on real data is essential.

Our first challenge is to deal with the significant variations presented in time-series data across different datasets or domains. These variations encompass differences in data distribution, time scales, signal-to-noise ratio, and feature types, among others. In order to establish a universally applicable approach for diverse time-series data, we draw inspiration from a classic method for general time series frequency domain extraction: the Fourier transform. Usually, in the field of time series analysis, widely used transformations such as the Discrete Fourier Transform (DFT) and Discrete Cosine Transform (DCT) are applied, while the Continuous Fourier Transform (CFT) is used in special cases. They all come with equations suggesting that each sample point contains information about the frequency domain Zieliński (2021). Therefore, CFT, DFT and DCT transformations convert these sample points into corresponding frequency components in the frequency domain, based on their underlying principle that each sample point in the time-series data can be represented by a set of frequency amplitude and phase. Building upon the fundamental capabilities of the frequency domain transformations, we have designed a method to synthesize data by superimposing several sine waves with varying phases, frequencies and amplitudes, simulating a range of rhythmic signals that may occur in the real world. Similar approaches to modeling rhythmic data have also been employed in other works Kükrer & İnce (2023) Maric (2017) Huo et al. (2021).

As the second challenge, real-world time-series data often contain noises of various frequencies and distributions, which can interfere with the accurate extraction of frequency components by the transformations. The transformations assume that each sample point can be represented exclusively by a combination of frequency, amplitude, and phase. However, in real data, individual signal sample points often incorporate non-rhythmic noise components that should not be converted into frequency, amplitude, and phase values. This inherent noise contamination impedes these transformations from precisely extracting the rhythmic portion's spectral information from the actual data. Secondly, time-series data may contain long-term features that remain undetected in the sampled data due to their minimum feature periods exceeding the sampling window's duration. As a result, these transformations may fail to identify such long-period features within the sampled data, potentially causing information loss or distortion in the frequency domain. In conclusion, although these transformations are useful in certain scenarios, their limitations are difficult to avoid when dealing with complex real-world time-series data.

Our approach to handling these challenges involves the explicit design of separately contained multi-source rhythmic data (MRD) information, various types of noise and their respective noise ratios (TN & NR), as well as trend information (TI) that extends beyond the sampling window. Real-world data typically lacks these explicit information. To address this, we develop a data synthesis approach that revolves around synthesizing MRD, TN & NR, and TI to create synthetic data with explicit information. Each set of synthetic data inherently corresponds to a specific set of generating parameters, including MRD, TN & NR, and TI. The data synthesis process only require sampling methods customized for MRD, TN & NR, and TI, with various random synthetic parameters. It does not rely on any learnable parameters to generate highly versatile synthetic data to solve the generalization problem of time-series data. This versatility is demonstrated by the fact that a DL model trained solely on InfoBoost's non-DL synthetic data outperforms that trained on large amounts of real data when validated on real data test sets.

In addition to applying synthetic data in unsupervised and self-supervised training, as an application of our synthetic data, we explored the idea of using the Rhythm, Noise, and Trend components in the synthesis process as reverse engineering to train a feature decomposer. This decomposer would be able to separate real-world time-series data into its Rhythm, Noise, and Trend constituents. We achieved this by training the decomposer with synthetic data as inputs and their corresponding Rhythm, Noise, and Trend components as labels. The resulting feature decomposer can effectively

divide any single-channel time-series into its Rhythm, Noise, and Trend parts. Given the decomposer's ability to isolate the rhythm component, we compare the frequency-domain representations derived from the raw data and the isolated rhythm. Visual analysis shows that the frequency-domain characteristics of the rhythm component are more distinct.

Summarized below are the main contributions of this work:

1. We present a general-purpose, non-DL approach (no need for real data to train) for synthesizing time-series data aimed at alleviating the challenges associated with acquiring domain-specific real time-series datasets.

2. We have validated the efficacy of our data synthesis method across multiple domains, such as finance, health, weather, among others. In most cases, training outcomes with synthesized data outperformed those using real data.

3. We enable the learning of a feature decomposer that only relies on synthetic data, enabling the decomposing of rhythmic, noise, and trend components of real time-series data.

## 2 RELATED WORKS

Although there are a variety of DL-based time-series data augmentation and synthesis methods Luo et al. (2023); Yang et al. (2022); Yang & Hong (2022); Dooley et al. (2023a), it is almost impossible to find a method that does not rely on training DL with real data and can be universally applied across domains, while simultaneously contributing downstream machine learning tasks Trirat et al. (2024). Expanding the probability distribution units for generated data using DL-based methods makes it challenging to ensure that the generated data covers unseen or diverse data distribution from other domains, even Meta-learning methods makes assumptions about tasks coming from the same distribution Swan et al. (2022). The development and exploration of non-DL-based universal time-series data synthesis methods provide a promising way to address the limitations of current DL-based approaches and improve the generalizability of synthetic data across diverse domains. FractalDB Kataoka et al. (2022) is known as a similar method in the field of imaging and $\pi$-GNN Yin et al. (2023a) in the field of graph.

In the field of time series, ForecastPFNDooley et al. (2023b) and ChronosAnsari et al. (2024) may appear similar to our work. However, these two approaches focus on implementing a zero-to-many training method using existing, straightforward data synthesis techniques paired with their designed time series prediction models. In contrast, our work concentrates on developing an innovative data synthesis method that benefits various model architectures and tasks. But, of course, to meet the requirements for comparison with related work, we have also detailed in Appendix subsection A.8 the fundamental differences in the data synthesis methods used by these two studies and model prediction performance under the same datasets and experimental settings.

Notably, our method enables models trained without real data to outperform those trained on real data in time-series field across all tested datasets, as demonstrated in subsection 4.1, even in the absence of real data or any real data information. Currently, data synthesis methods that claim to achieve zero to many require specific information from the dataset. For example, ForecastPFN needs the temporal periodicity (such as year, month, day) of the dataset, whereas Chronos's data synthesis method, which relies 90% on TsMix, requires segments of real data for concatenation and synthesis.

Furthermore, our work innovatively enables unconditional feature decomposition(without limitations on temporal periodicity or other additional information) based on DL for trends, noise, and rhythms features using our synthesized data as train data. Classic methods like STL decomposition require accurate timepoint index information from the dataset and need extra predefined parameters for seasonality and trend to function effectively. The feature decomposition is demonstrated in subsection 4.3.

## 3 METHODOLOGY

### 3.1 INFOBOOST SYNTHETIC TIME-SERIES DATA

In this section, we will demonstrate how to generate multi-source rhythmic data (MRD), different types of noise and their noise ratios (TN & NR), and trend information (TI) based on parametric design. These components will be combined according to their respective ratios to create synthetic data. Ablation study for 3 components is illustrated in appendix subsection 4.4.

### 3.1.1 GENERATING MULTI-SOURCE RHYTHMIC DATA

To synthesize rhythmic data that accurately reflects the diversity and complexity of real-world time series, we follow the Nyquist-Shannon Sampling Theorem to ensure that all sine waves' frequencies are uniformly sampled during the synthesis of rhythms. Our design involves creating rhythmic data comprising both simple and complex waveforms, which are constructed from a random number of sine waves with uniformly random amplitudes, frequencies, and phases.

Motivation and Theory: 1)Frequency Distribution: Each sine wave represents a distinct frequency component that simulates a specific rhythm in real-world data. By adhering to the Nyquist-Shannon Sampling Theorem and using uniform sampling for the frequencies, we ensure that each discrete sample has an equal probability of representing the entire spectrum from the minimum to the maximum frequency. This approach avoids potential biases introduced by concentrating frequencies around certain values, leading to more realistic synthetic data. 2)Amplitude Normalization: For each sine wave, we assign a random initial weight sampled uniformly from the range [0, 1]. These weights are then normalized, ensuring that the resulting amplitudes are scaled between 0 and 1. This step guarantees that all sine waves, regardless of their frequencies, have an equal chance of becoming the dominant rhythm feature in the final composite signal. 3)Phase Parameter: Although often neglected in machine learning applications, phase variations significantly impact the shape of the composite signal even when the frequencies and amplitudes remain constant. Uniformly sampling the phase ensures that the phase distribution does not introduce additional bias into the synthetic data, allowing for a wide range of possible signal shapes. The phase values typically range between 0 and $2\pi$, aligning with the conventional understanding of the sine function's periodicity.

Frequency Selection: As shown in Equation 1, 1)Upper Limit Frequency ($f_{\max}$): The upper limit frequency is set to half the sampling frequency ($\frac{f_s}{2}$), where the sampling frequency ($f_s$) is defined as the inverse of the time interval between adjacent discrete time points. This ensures that all sine waves are sampled without aliasing, adhering to the Nyquist-Shannon Sampling Theorem. 2)Lower Limit Frequency ($f_{\min}$): The lower limit frequency is approximately the inverse of the total number of discrete time points (N) in the sampling window. This ensures that the lowest frequency sine wave can complete at least one full cycle within the sampling window.

$$f_{\max} = \frac{f_s}{2} = \frac{1}{2t} \quad , \quad f_{\min} \approx \frac{1}{N}. \tag{1}$$

The synthesis of multi-source rhythmic data is presented in Figure 2.

### 3.1.2 GENERATING DIFFERENT TYPES OF NOISE

To simulate noise for all types of time-series data, we designed a synthetic noise generator that encompasses 15 different types of noise distributions, classified into 5 primary categories. This design offers a wide range of noise distributions, and closely mirror real-world scenarios. To achieve this, for discrete distributions, we adopted Bernoulli Sinharay (2010), geometric Sinharay (2010), and Poisson distributions Sinharay (2010). Additionally, included heavy-tailed distributions Kotz et al. (2001), distributions related to the normal distribution (t-distribution Li & Nadarajah (2020) and Pareto distribution Coles (2001)), shape parameter distributions (Beta and Gamma distributions Liu & Serota (2023)), scale parameter distributions (exponential family distribution Gupta et al. (2010) ) and normal distribution family Wiley & Wiley (2020). 15 noise distributions and their 5 categories are listed in Figure3.

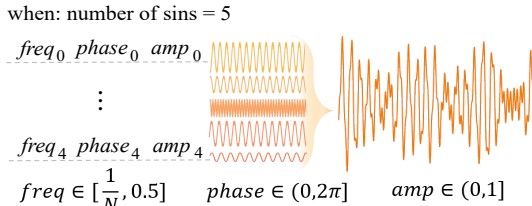

when: number of sins = 5

$freq_0$ $phase_0$ $amp_0$

$\vdots$

$freq_4$ $phase_4$ $amp_4$

$freq \in [\frac{1}{N}, 0.5]$     $phase \in (0,2\pi]$     $amp \in (0,1]$

Figure 2: This image illustrates a possible set of five corresponding sine waves, each obtained by random sampling of frequency phases and amplitudes within their respective ranges, and it should be noted that the number of sine waves is also randomly determined. Additionally, the image showcases the composite rhythmic data generated by the superposition of these randomly determined sine waves.

| Noise Name | Category | Noise Name | Category |
|---|---|---|---|
| Normal | CCD | Pareto | HTD |
| Student's t | CCD | Generalized Gamma | HTD |
| Uniform | CCD | Log-Normal | DRND |
| Exponential | CCD | Exponential LogNorm | DRND |
| Poisson | CDD | Gamma | SPD |
| Binomial | CDD | Beta | SPD |
| Negative Binomial | CDD | Weibull | SPD |
| | | Rayleigh | SPD |

Figure 3: The list of noise distributions along with their corresponding categories. The "Category" column specifies the category each noise type belongs to. The categories are abbreviated as follows: CCD (Common Continuous Distributions), CDD (Common Discrete Distributions), HTD (Heavy-Tailed Distributions), DRND (Distributions Related to Normal Distribution), and SPD (Shape Parameter Distributions).

Due to the highly uncertain nature of noise distributions in real data, when sampling the noise, we undertake the following three steps:

1. We customize random parameter sampling based on the fundamental parameters of each noise distribution to introduce relative randomness into each distribution, if $X$ is the Sampled noise, the equation as follow:

$$X \sim NoiseDist(p). \tag{2}$$

2. We perform a partial y-axis inversion on the sampled results of each parameter distribution. This ensures that distributions overly concentrated around a maximum or minimum value do not adversely impact the uniformly normalized sampled results.

3. We apply random kernel size smoothing to the sampled noise, enhancing the diversity of the noise data distribution and simulating temporal dynamics that are difficult to classify as rhythmic information in some real datasets. To prevent overlap between noise and trend information, the kernel size for the random smoothing in the third step is constrained to a relatively small proportion compared to that of the total data length. When Kernel size:$K = 2k + 1$ and Smoothed noise: $Y_t$ and $X$ stands for Sampled noise:

$$Y_t = \frac{1}{2k+1} \sum_{j=-k}^{k} X_{t+j}. \tag{3}$$

### 3.1.3 GENERATING TREND INFORMATION

Generating trend information involves a randomized selection between two distinct methods. This random selection process providing a degree of stochasticity in simulating different types of data trends. Our trend generation process involves the random selection between two distinct methods:

1. Multi-Sine Trend Generation: This method enables the simulation of complex periodic patterns by generating multiple sine waves with random parameters and combining them to form a composite trend. we utilize a similar approach to the one used in subsubsection 3.1.1, which entails the super-position of multiple sine waves. However, unlike subsubsection 3.1.1, when generating long-period trends, we constrain the superposition of sine waves to ensure a relatively smaller number of spikes and less complex waveforms in the resulting trend. We ensure that the generated minimum period is greater than the range captured by the sample window. Additionally, we introduce a random multiplier greater than 1 to enhance the diversity of the generated data. This adjustment ensures that the generated periods exhibit a certain level of diversity and can simulate a variety of periodic trends across a wider range. If $A_i$ is the amplitude of $i_t h$ sine wave , $f_i$: frequency and $\phi_i$: phase

$$T(t) = \sum_{i=1}^{N} A_i \sin(2\pi f_i t + \phi_i).$$
(4)

2. Random Noise Trend Generation: This method allows for the simulation of random fluctuations or irregularities often observed in real data, and introduces controlled randomness and smooths out the generated trend, replicating the stochastic nature of many real-world trends. By generating noise with the same method as described in subsubsection 3.1.2, and applying a larger kernel size for smoothing, this approach enhances the diversity and stability of the data, mitigates overfitting, and better simulates real-world data trends. When $N(t)$ stands for noise signal sampled from random noise distribution, to prevent overlap between noise and trend information, the kernel size for the random smoothing in the trend is constrained to be relatively big proportion compared to noise smooth window, in the same equation described in Equation 3.

The random selection between these two methods aims to enhance the diversity and stochasticity of the generated trend data, providing a more realistic features of the multifaceted nature of data trends commonly observed in real-world datasets.

### 3.1.4 SIGNAL-TO-NOISE-AND-TREND RATIO

Given the rhythm noise and trend information generated through their respective random parameters, the next step involves standardizing each of the three generated outputs to fall within the range of -1 and 1. This standardization facilitates the computation of the contribution ratio of each synthetic component in the final composite data. As this ratio directly determines the signal-to-noise ratio of the rhythmic information in the composite data, it significantly influences the overall performance and characteristics of the synthesized data. We will randomly generate a set of three ratios, whose sum is 1, to serve as the ratios for the rhythmic, noise, and trend components:

$$r_{\text{rhyth}} + r_{\text{noise}} + r_{\text{trend}} = 1.$$
(5)

Consequently, we will utilize these individual ratios to weight the generation of the final composite data, combining the components based on their respective ratios:

$$\text{Sync} = r_{\text{rhyth}} \times \text{Rhyth} + r_{\text{noise}} \times \text{Noise} + r_{\text{trend}} \times \text{Trend}.$$
(6)

### 3.2 UNIVERSAL UNCONDITIONAL TIME SERIES FEATURES EXTRACTION

As an unique application of our synthetic data, we will demonstrate how to train a features extractor solely based on the synthetic data generated from the random parameters and the composite data obtained from subsection 3.1. This features extractor is designed to explicitly separate MRD, TN & NR, and TI (rhythmic, noise, trend)information, using only the synthetic data for training. Classic methods like STL decomposition require accurate timepoint index information from the dataset and need extra pre-defined parameters for seasonality and trend to function effectively, our method only needs input data.

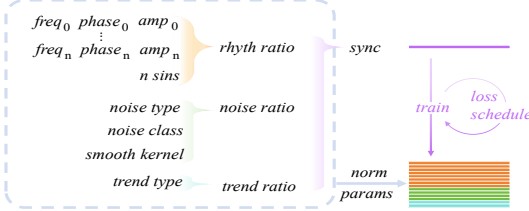

Figure 4: Normalization of parameters (norm params) used in the data synthesis process. The normalized parameters are organized into a multi-channel matrix, aligning with the sampling window length of the synthetic data.

In data synthesis process, the majority of parameters consist of continuous values, such as frequencies, phases, amplitudes, and ratios. However, there are also crucial parameters that are composed of discrete information, including the number of sine waves, noise type and class, trend class, and similar kernel size parameters resembling smoothing windows, whose actual values may scale significantly with the length of the data sampling window.

To ensure that a wide range of values and diverse parameter types numbering over a dozen (the total count being dependent on the preset range of the number of sine waves, typically empirically set between 3 and 10), can be effectively fitted as labels for deep models and to mitigate potential interference from different types of loss functions, we have custom-tailored a standardization scheme for all parameters. Ultimately, all values are constrained within the range of -1 to 1 (with some parameters set between 0 and 1). Through broadcasting or interpolation, we map all parameters to a length equivalent to the sampling window of the synthetic data. This process ensures that MRD, TN & NR, and TI, along with the generated parameters, are organized into a multi-channel matrix of the total parameter count multiplied by the sampling window length, serving as labels. The synthetic data is then utilized as input to train the features extractor. The parameters contained within the multi-channel normalized parameters are shown in Figure 4. For normalization, if $\hat{P}ij$ stands for normalized $i_t h$ parameter value at the $j_t h$ data point and $P_{ij}$ stands for original parameter value.

$$\hat{P}ij = \begin{cases} \frac{Pij - \min(P)}{\max(P) - \min(P)} \times 2 - 1, & \text{for } P \text{ in } [-1, 1]; \\ frac{P_{ij}} - \min(P)\max(P) - \min(P), & \text{for } P \text{ in } [0, 1]. \end{cases} \tag{7}$$

The training details are presented in appendix subsection A.7

## 4 EXPERIMENTS

To test the generalizability of the synthetic data within the InfoBoost framework and evaluate the features extraction performance of the features Extractor on real-world data, we gathered 35 publicly available time-series datasets from two prominent time-series collections, the Tslib and Monash collections Wu et al. (2023); Godahewa et al. (2021), along with various other datasets, all datasets names presented in appendix subsection A.9. These datasets encompass a wide range of data types, including electroencephalography (EEG) data, epidemiological data, electricity data, cryptocurrency data, traffic data, and meteorological data. Due to space limitations, we have placed the imputation experiments(subsection A.6) and comparisons with related work(subsection A.8) and second ablation study(subsection A.3) in the appendix.

### 4.1 SYNTHETIC DATA COVERAGE VERIFICATION VIA UNSUPERVISED LEARNING

Setting: Unsupervised autoencoding task requires the model to learn from input data and recreate the original data, and evaluate the model's learning performance to learn data features and generalize to new, unseen data, without the necessity of labels or domain-specific inferences. In the 'sync' group, we train using only synthetic data but test using real data, the goal is to determine if the synthetic data sufficiently covers the range of real-world data features.

Training: With MSE loss, We trained models on InfoBoost's synthetic data and random real-world subsets for unsupervised-autoencoding. The rest of the real datasets served as a test to compare

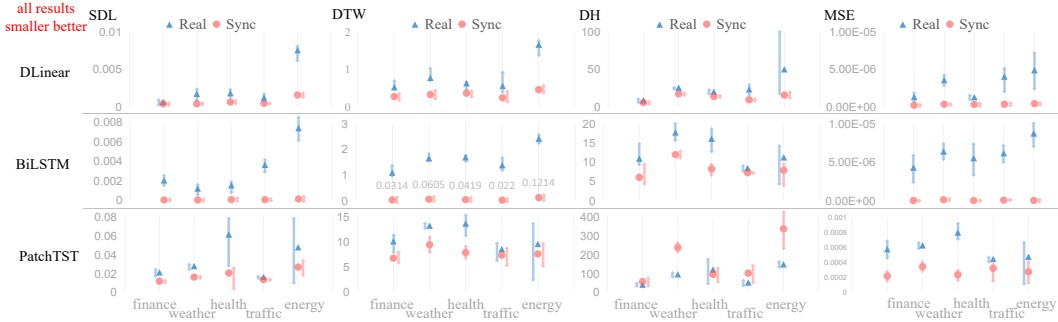

Figure 5: The experimental results in Figure 4.1 show that smaller values for all metrics indicate better performance. Unsupervised autoencoding performance directly across various domains of time-series data(categorized into five major groups).

synthetic-trained models with real-data-trained ones. To ensure fairness, approximately 70% of the real-world datasets, specifically 24 datasets, were randomly selected as the baseline training data. Due to the entirely random selection, it is difficult to avoid overlaps between some of the baseline training datasets and the test set in terms of data types, further increasing the difficulty of surpassing the baseline. The selected baseline training set ultimately comprised a specific number (usually around 200,000 segments depending on random selection) of instances of real time-series data. To put it into perspective using the common methodology employed in time series studies, the number of time points used in the training data for this study amounts to approximately 160,000,000 time points. Similarly, our synthetic data generation also yielded a similar number of instances of synthetic time-series data, which were used to train the model for Unsupervised autoencoding based on InfoBoost's synthetic data.

Evaluation matrices: We evaluated four distinct losses to measure the unsupervised autoencoding performance. 1) Structural dissimilarity loss, derived by subtracting the structural similarity index (SSIM) Venkataramanan et al. (2021) from 1, is to assess structural variance between the reconstructed and original data. 2) Dynamic time warping (DTW) Salvador & Chan (2007) distance, a metric suitable for measuring similarity between two time series, accomodates time shifts and stretches. 3) Distance between histograms Cha & Srihari (2002) is to gauge dissimilarities between the histograms of the reconstructed and original time-series data, providing insights into their similarity. 4) Mean squared error (MSE) loss is a common metric used to quantify differences between the reconstructed and original data. By incorporating 4 diverse losses, we comprehensively evaluated Unsupervised autoencoding performance from various perspectives. To mitigate the impact of model architecture selection on experimental results, we opted for three highly representative diverse model structures: recurrent architecture BiLSTM Abduljabbar et al. (2021), linear networks DLinear Zeng et al. (2023), and transformer architecture PatchTST Nie et al. (2023). These were chosen to assess the Unsupervised autoencoding performance on a real data test set after training solely on synthetic data and real data, respectively. The results are shown in Figure 5.

The results: Figure 5 indicate that, except for two scenarios (such as the DH & DTW value of PatchTST on the Energy class dataset), the performance of the model trained on InfoBoost's synthetic data surpasses that of the model trained on real data in the real data test set in 55 out of 60 testing scenarios. It demonstrates the generalization capability of InfoBoost's synthetic data across various types of scenarios. Even in other testing scenarios of the Energy class dataset, although slightly inferior to the performance after training on real data, the performance of the model trained on synthetic data is mostly very close. The reasons for the slightly inferior performance of synthetic data in the PatchTST scenario on the Energy class dataset will be discussed in Figure A.1.

## 4.2 SELF-SUPERVISED DOMAIN-SPECIFIC FORECASTING

Setting: To validate the effectiveness of InfoBoost's synthetic data in assisting models on common time-series tasks across various real-world scenarios, we employ a self-supervised forecasting task and utilize the same model across five distinct domains. Within each domain, we conduct training and testing based on both real and synthesized data, using that domain's unique dataset. We chose the same models as in Unsupervised experiment subsection 4.1.

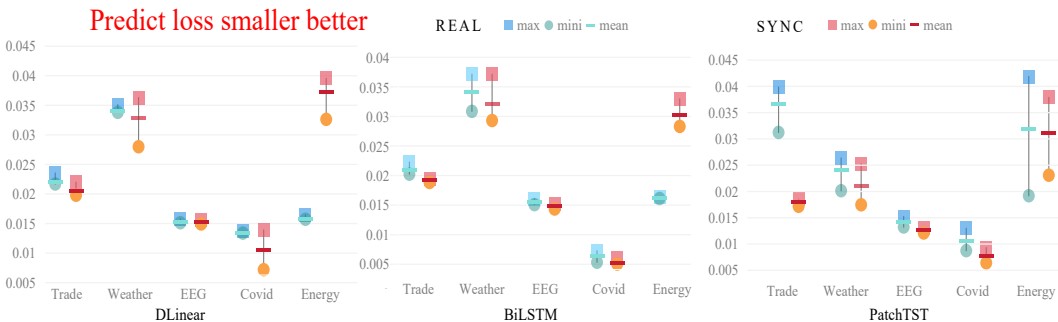

Figure 6: Forecasting experiments per domain, lower losses the better. The numerical version illustrate in Table 4.

Training: Specifically, we allocate the last one-eighth portion of each dataset as labels and the preceding seven-eighth segment as inputs for our training and evaluation. Furthermore, approximately two-thirds of the data from each domain are selected for the training set (serving as a proxy for the limited datasets that can be collected in real scenarios), while the remaining one-third is designated for the test (acting as a substitute for the unseen data that is difficult to obtain for training purposes), training with MSE loss. In this experiment, each domain is trained separately.

Evaluation: We assess the models' performance by calculating the Mean Squared Error (MSE) loss between the model's predicted patches for the last one-eighth segment and their corresponding actual patches in the real data. The outcomes of multiple experiment runs per domain are summarized with the maximum, minimum, and average losses depicted in Figure 6, differentiating between models trained on synthetic data and those trained on real data. This visualization allows for a clear comparison of how InfoBoost's synthetic data impacts model performance relative to training solely on authentic datasets across all domains.

Results: In the five prevalent time-series domains including Trade, Weather, EEG, Covid, and Energy, the DLinear model trained exclusively on synthetic data consistently demonstrated superior mean loss and minimal loss performances. Notably, the only exception was observed in the Energy domain, where it struggled to match the results achieved through training on real data. This finding aligns with the experimental outcomes presented in Figure 5, reinforcing the notion that InfoBoost's synthetic data effectively empowers tested models to transcend the limitations of real data reliance. Consequently, except for the Energy sector, these models exhibit enhanced performance on unseen data within these scenarios, highlighting InfoBoost's capability to facilitate improved generalizability across diverse time-series applications.

## 4.3 CASE STUDY OF EXPLICIT FEATURE EXTRACTION

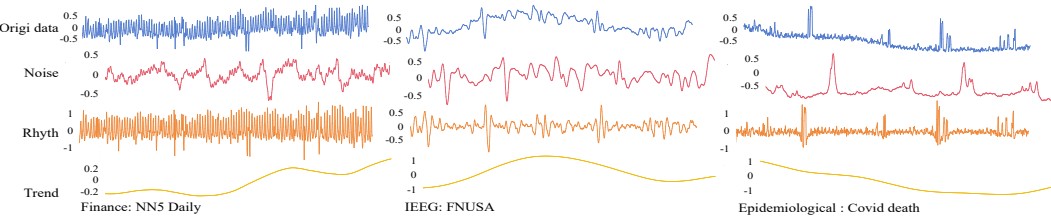

Figure 7: The general feature extractor extracts MRD, TN, NR, and TI information from three distinct and characteristic time-series datasets, following training solely on synthetic data. The ratio information has already been incorporated in a weighted manner across the data scales of each extracted feature after decomposition.

This section will showcase, through visualizations, the performance of a universal features extractor trained on synthetic data containing MRD, TN & NR, and TI information. The visualizations will demonstrate the extraction efficacy of MRD, TN & NR, and TI across various data types, which

is shown in Figure 7. Due to the influence of noise and trend information, the frequency domain information extracted by the commonly employed DFT method in DL often experiences a reduction in quality. Therefore, in Figure 9, we present the DFT frequency domain extraction results of rhythmic information based on the InfoBoost feature extractor. This demonstrates that the disrupted frequency domain information in the original data's DFT results can be removed for rhythmic information extraction in the frequency domain.

This section visually demonstrates the fundamental functionality of the feature decomposer by presenting the original forms of data instances from three domains alongside the Rhythms, Noise, and Trends (as depicted in Figure 7) extracted using the feature decomposer trained on synthetic data produced by InfoBoost.

### 4.4 ABLATION OF RHYTH, NOISE AND TREND COMPONENTS

Setting & Evaluation: The ablation experiment aimed to validate the contributions of the three components—Rhyth, Noise and Trend—of the synthetic data to the model's performance. All models were trained exclusively on the synthetic data and subsequently subjected to unsupervised autoencoding on all real datasets. We used the DLinear modelZeng et al. (2023) as a baseline, which has a relatively simple architecture and demonstrated median performance in Experiment Figure 5. Performance differences across various configurations were evaluated by computing the Mean Squared Error (MSE) between the original real data and the model's output.

Results: The final results indicated that the model achieved the best performance when all three modules were included in the synthetic data, as evidenced by the loss values in Table 1. According to the experimental results, the configuration RNT, which incorporates all three rhythmic, noise, trend components for data synthesis, yields the lowest MSE, indicating that the quality of the synthesized data is optimal when all three components are used concurrently.

| Sync Config | Min MSE | Max MSE | Mean MSE |
|---|---|---|---|
| RNT | $1.5 \times 10^{-7}$ | $1.6 \times 10^{-7}$ | $1.5 \times 10^{-7}$ |
| NT | $3.7 \times 10^{-7}$ | $3.9 \times 10^{-7}$ | $3.8 \times 10^{-7}$ |
| RT | $2.0 \times 10^{-7}$ | $2.1 \times 10^{-7}$ | $2.1 \times 10^{-7}$ |
| RN | $4.1 \times 10^{-7}$ | $4.2 \times 10^{-7}$ | $4.8 \times 10^{-7}$ |

Table 1: Ablation Experiment Results: Mean Squared Error (MSE) for Different Synthetic Data Configurations. We employ the abbreviations R, N, and T to represent the components Rhyth, Noise, and Trend, respectively.

## 5 CONCLUSION

In this study, we have introduced a unique approach, marking the first to simultaneously fulfill the requirements of a universal time-series data synthesis method that does not rely on real data or DL, and a universal time-series data feature decomposition and extraction method that does not require fine-tuning on real data. Most notably, our method empowers models trained in the absence of real data information to outperform those trained on real data across almost all tested datasets. This achievement opens up a new path for future time-series data analysis and modeling, as well as a new solution for time-series unsupervised or self-supervised learning.

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

# A APPENDIX

## A.1 UNSUPERVISED LEARNING WITH UNLIMITED QUANTITY SYNTHETIC DATA

While the model based on synthetic data outperforms the model trained on real data, we observe a decline in performance with excessive epochs at a fixed learning rate, particularly in transformer-based

|  | METHOD | MINVALILOSS | MINVALILOSS EPOCH |
|---|---|---|---|
| DLINEAR | REAL | $4.0 \times 10^{-7}$ | 21 |
|  | SYNC | $1.5 \times 10^{-7}$ | 21 |
|  | UNLIMITSYNC | $8 \times 10^{-8}$ | 8 |
|  | METHOD | MINVALILOSS | MINVALILOSS EPOCH |
| BILSTM | REAL | $7.4 \times 10^{-7}$ | 68 |
|  | SYNC | $3 \times 10^{-8}$ | 65 |
|  | UNLIMITSYNC | $1 \times 10^{-8}$ | 30 |
|  | METHOD | MINVALILOSS | MINVALILOSS EPOCH |
| PATCHTST | REAL | $8.70 \times 10^{-6}$ | 21 |
|  | SYNC | $7.52 \times 10^{-6}$ | 5 |
|  | UNLIMITSYNC | $7.26 \times 10^{-6}$ | 4 |

Table 2: Table for subsection A.1.The minimum validation loss for the three test models are presented, each of which is trained on different data sets and minimizes on different epochs. The data in the table indicates that by replacing the training data with new synthetic data at every epoch, the models can achieve better performance in fewer epochs.

models PatchTST. This suggests that although synthetic data enhances generalizability, limiting the size of the training set for the sake of fairness in the experiment may cause the model to overfit to specific features within this subset.

To validate this hypothesis, we modified the Unsupervised autoencoding task in Experiment Figure 4.1 to remove the limit on synthetic data quantity. For each completed epoch, a new set of synthetic data was generated to serve as the training data for the Unsupervised autoencoding task. We then compared the change in validation set loss for each epoch between the limited synthetic data Unsupervised autoencoding task and the unlimited synthetic data Unsupervised autoencoding task, using the same model architectures. Based on the experimental results shown in Figure 8 and Table 2, models trained with fixed learning rates and synthetic data replacement at every epoch demonstrates the ability to rapidly and fit well to the validation set with very few epochs, outperforming both the model trained on real data and the model trained on a limited set of synthetic data. Furthermore, as the number of epochs increases, there is no significant increase in validation loss. Instead, the fluctuation in validation loss corresponds to the variation in the generated training data.

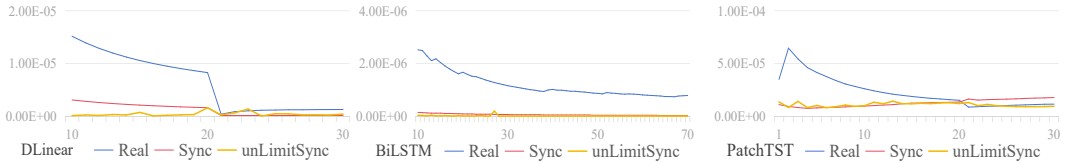

Figure 8: Figure for subsection A.1. The comparative results, depict the change in the validation set (composed of randomly selected real data) loss for each epoch. Unlike 'Sync', 'unlimitSync' represents the generation of an entirely new synthetic dataset for training at each epoch.

## A.2 LIMITATIONS AND FUTURE WORK

Given that our method is designed specifically to synthesize Rhythms, Noise, and Trends, it is highly likely that extreme events appearing in non-rhythmic formats(For example, in weather forecasting, there are always unpredictable variables evolving in the form of time series) would be captured as part of the Noise component when utilizing a feature decomposer trained on InfoBoost synthetic data, as exemplified in subsection 4.3. Furthermore, extreme events that exhibit exceptionally mild characteristics within the sampling window might also be treated as part of the Trend component. Undoubtedly, addressing the handling of such extreme events constitutes a promising avenue for future work in this work.

## A.3 ABLATION EXPERIMENT FOR THE RANDOM SMOOTHING OF NOISE IN STEP3 OF SUBSUBSECTION 3.1.2 GENERATING DIFFERENT TYPES OF NOISE

In machine learning and AI, noise is often seen just as a way to represent a distribution, so we don't usually look at how noise changes over time(such as the density of spikes in the sampling results or the overall trends of growth or decline). However, when we create noise for our studies, we must consider that the impact of noise on data in real-world scenarios goes beyond merely disturbing the data distribution; it also encompasses various types of impacts on the temporal features of rhythmic data. To account for this, we add a smoothing effect with a randomly sized kernel. This changes how the noise looks over time and could affect the rhythmic patterns in the data differently. This experiment tests if this random smoothing makes the synthetic data better.

We will use the same unsupervised setting as in Appendix A.3 to compare the performance of models trained on synthetic data with and without random smoothing of the noise component on real vali-data.

Table 3: Comparison of model performance with and without random smoothing of the noise component on real validation data

|  | min MSE | max MSE | mean MSE |
| --- | --- | --- | --- |
| Noise part with random smooth | $1.5 \times 10^{-7}$ | $1.6 \times 10^{-7}$ | $1.5 \times 10^{-7}$ |
| Noise part no smooth | $4.7 \times 10^{-7}$ | $5.0 \times 10^{-7}$ | $5.1 \times 10^{-7}$ |

## A.4 NUMERICAL RESULTS FOR SUBSECTION 4.2 SELF-SUPERVISED DOMAIN-SPECIFIC PREDICTION

To clarify the results more clearly, Table 4 here are the numerical versions of the prediction experiment results described in Figure 6:

Table 4: Forecasting experiments per domain, lower losses the better.

| Model | Dataset | Mean ± SD | |
| --- | --- | --- | --- |
| | | Real | Sync |
| DLinear | Trade | $0.02207 \pm 0.0045$ | $0.02056 \pm 0.0013$ |
| | Weather | $0.03408 \pm 0.0062$ | $0.03289 \pm 0.0083$ |
| | EEG | $0.01532 \pm 0.0006$ | $0.01522 \pm 0.0014$ |
| | Covid | $0.01347 \pm 0.0003$ | $0.01057 \pm 0.0067$ |
| | Energy | $0.01589 \pm 0.0006$ | $0.03723 \pm 0.0069$ |
| BiLSTM | Trade | $0.02101 \pm 0.0012$ | $0.01921 \pm 0.0011$ |
| | Weather | $0.03405 \pm 0.0062$ | $0.03228 \pm 0.0078$ |
| | EEG | $0.01553 \pm 0.0004$ | $0.01489 \pm 0.0003$ |
| | Covid | $0.00640 \pm 0.0019$ | $0.00522 \pm 0.0010$ |
| | Energy | $0.01624 \pm 0.0001$ | $0.03039 \pm 0.0046$ |
| PatchTST | Trade | $0.03664 \pm 0.0086$ | $0.01793 \pm 0.0033$ |
| | Weather | $0.02400 \pm 0.0062$ | $0.02100 \pm 0.0042$ |
| | EEG | $0.01421 \pm 0.0010$ | $0.01256 \pm 0.0047$ |
| | Covid | $0.01051 \pm 0.0022$ | $0.00773 \pm 0.0025$ |
| | Energy | $0.03197 \pm 0.0227$ | $0.03109 \pm 0.0049$ |

## A.5 USAGE OF EXPLICIT FEATURE EXTRACTION

Moreover, Figure 9 showcases a clearer visual discrimination by presenting the decomposed Rhythmic component compared to the frequency domain calculation results from the original data. Collectively, these illustrations serve to demonstrate one application of the InfoBoost synthetic data in

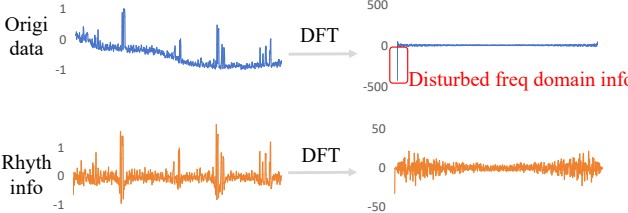

Figure 9: DFT results in DL tasks are based on the input depicted in Figure 7, the COVID death data. Results exhibit an disruption in data's DFT result. After extracting only the rhythmic information, the DFT results for the rhythmic data exhibit higher quality in the frequency domain.

facilitating such decomposition processes, also, it substantiates the fundamental feasibility of training a feature decomposer based on synthetic data.

### A.6 Imputation task only trained with synthesis data

Setting:We train DLinear, BiLSTM, PatchTST on real and sync data to compare their impu perform. We apply random (uniform) segment masking, masking 10-30% of the data length. Masked data as input, original data as target. Loss is computed using MSE between output and original data.

Training:Real data training uses 237,269 data segments and the test set contains 91,055. Experiments on single 3090 GPU. Synthetic data training with 237,240 synthetic segments and the same real test data. Batch sizes are maximized based on each model's VRAM usage.

Results are presented as MSE losses between the test set's complete data and the model's imputed output, lower values indicating better performance. The models trained on our synthetic data outperform those trained on real data in imputation. This suggests that the sync data's probability distribution is more robust and generalizable than training with real data:

Table 5: MSE Losses for Imputation Task Using Synthetic vs. Real Data(Smaller better)

| Model | Real | Sync |
|---|---|---|
| DLinear | $6.302 \times 10^{-3} \pm 1.027 \times 10^{-4}$ | $2.527 \times 10^{-3} \pm 1.057 \times 10^{-4}$ |
| BiLSTM | $7.911 \times 10^{-4} \pm 4.971 \times 10^{-5}$ | $4.646 \times 10^{-4} \pm 3.325 \times 10^{-5}$ |
| PatchTST | $3.923 \times 10^{-3} \pm 3.937 \times 10^{-4}$ | $0.889 \times 10^{-3} \pm 1.112 \times 10^{-4}$ |

### A.7 Features Extraction Training

After obtaining the multi-channel normalized parameters (norm params) matrix, which encapsulates the majority of information constituting the synthetic data without containing the data itself, we can train a features extractor solely on the task of learning from the synthetic data input and the norm params as labels. The norm params encompass various dimensions such as MRD, TN & NR, and TI, are derived from random sampling and contain diverse continuous random functions. Consequently, the combination of parameters used to generate synthetic data is not limited in quantity, allowing for an infinite variety of corresponding synthetic data. Therefore, training a deep model on the synthetic data to learn the task of MRD, TN & NR, and TI from norm params makes it nearly impossible for the model to overfit. To achieve optimal extraction of MRD, TN & NR, and TI within a reasonable timeframe, we adopt a concise linear loss schedule to train the features extractor. The training process is also depicted in Figure 4. The training loss is MSE loss function:

$$L(\Theta) = \sum_{i=1}^{N} \text{MSELoss}\left(f_\Theta(x_i), y_i\right) \tag{8}$$

During the training process, in our quest to find the most suitable model architecture for the extraction of MRD, TN & NR, and TI tasks, we experimented with various model architectures including Bi-LSTM Abduljabbar et al. (2021), DLinear Zeng et al. (2023), PatchTST Nie et al. (2023), among

others. Based on the visualization results, we selected DLinear as the InfoBoost's features Extractor due to its superior visual performance.

## A.8 COMPARISON WITH FORECASTPFN AND CHRONOS

In the field of time series data, the data synthesis methods used in ForecastPFNDooley et al. (2023b) and ChronosAnsari et al. (2024) may sound similar to ours at first glance. In fact, compared to theirs, our synthesis method has several key advantages:

1)Unlike ForecastPFN, our method does not have specific time scale (year, month, day) and data length restrictions. 2)Our rhythmic method is more diverse and complex compared to ForecastPFN's seasonal method, allowing for more varied and intricate rhythms without being limited to specific data scales or domains. 3)Our noise component includes five major categories with 15 different noise distributions, providing a more comprehensive simulation of real-world noises compared to ForecastPFN and KernelSynth (used in Chronos). 4)Our trend component, with its random selection of rhythm and noise types, offers a broader coverage of real-world scenarios compared to ForecastPFN's relatively simple design. 5)Unlike ForecastPFN, which requires information about real data scales, and TsMix in Chronos, which relies on 90% real data for synthesis, our method generates data without needing any information or segments from real data, yet achieves better performance on real data tasks.

In summary, our work, compared to the synthesis methods of ForecastPFN and Chronos (TsMix + KernelSynth), can be illustrated in the following table:

| Synthesis Method | Don't Need Real Data | Dozen Different Noise Dists | Don't Need Specific Time Periods | Flexible Data Length | Rhythm & Noise & Trend |
|---|---|---|---|---|---|
| InfoBoost(Ours) | ✓ | ✓ | ✓ | ✓ | ✓ |
| ForecastPFN | | | | | ✓ |
| TSMix | | | | ✓ | |
| KernelSynth | ✓ | | ✓ | ✓ | |

Table 6: Comparison of Synthesis Methods

And we tested the data synthesis methods of Chronos(TsMix + KSync) and our own data synthesis method according to the experimental setup and evaluation metrics of ForecastPFN.

Setting: We compared these with the best-performing experimental data provided in the original ForecastPFN paper(Results of FEDformer, ForecastPFN, Informer & SeasonalNaive). When testing TsMix + KSync and our InfoBoost synthetic data, we used the DLinear model as a baseline, which has a relatively simple architecture and was also utilized as a baseline in the original Chronos paper.

Data: In the seven datasets used for ForecastPFN, to access the PeMS datasets (traffic) from the Data Clearinghouse, it is necessary to have additional registration steps and to be approved; therefore, we conducted a direct comparison with the results reported in the ForecastPFN on the other datasets. The training parameters were set according to the Data Budget = 500 experiment group in the ForecastPFN paper: an input context length of 36 and an output prediction length ranging from 6 to 48.

And we observed that a significant proportion of the data segments in the ECL dataset consists of all-zero segments. Direct testing benchmarks showed much lower losses than those reported for ForecastPFN, but the paper does not detail how these all-zero segments in the ECL data were handled; thus, we removed the ECL from our results.

Results: As shown in Table 7, ours outperformed ForecastPFN and Chronos in datasets other than weather. For Chronos' performance in weather, we believe that since the single weather dataset includes 13 million data points(3 years data), the synthetic data generated using real data in Chronos's TsMix method can benefit more from the sufficient volume of real data. This result and inference further demonstrate that in scenarios where the data volume is not abundant, our synthetic data can bring about greater benefits.

Our assessment of InfoBoost's performance relative to ForecastPFN and TsMix+KSync (Chronos) in this experiment primarily attributes the difference to the coverage of probability distributions

| Model | ETTh1 | ETTh2 | Exchange | Illness | Weather |
|---|---|---|---|---|---|
| FEDformer | 0.133 | 0.352 | 0.068 | 0.707 | 0.188 |
| ForecastPFN | 0.127 | 0.33 | 0.058 | 1.091 | 0.009 |
| Informer | 0.144 | 0.253 | 0.529 | 4.394 | 0.224 |
| SeasonalNaive | 0.203 | 0.554 | 0.028 | 1.41 | 0.017 |
| Chronos(TsMix+KSync) | $0.146 \pm 0.002$ | $0.164 \pm 0.005$ | $0.015 \pm 0.0002$ | $1.00 \pm 0.004$ | $0.008 \pm 0.001$ |
| InfoBoost(Ours) | $0.099 \pm 0.005$ | $0.118 \pm 0.004$ | $0.014 \pm 0.001$ | $0.40 \pm 0.08$ | $0.010 \pm 0.0001$ |

Table 7: Comparison of Different Models on Various Datasets. The results of FEDformer, ForecastPFN, Informer, and SeasonalNaive are directly taken from the original ForecastPFN paper Dooley et al. (2023b).

within the train-set. This coverage is intuitively reflected in the size of the training set; when the dataset is limited in size, the train set can only expose the model to a limited diversity of features and a less comprehensive range of probability distributions. In Chronos, 90% of TsMix and 10% of KSync synthetic data are utilized, where TsMix requires real data for augmentation, thus heavily depending on the coverage of the dataset.

Here are the sizes of the five benchmark datasets included in ForecastPFN:

Illness: 966 data points,

Exchange: 7,588 data points,

ETT1 & ETT2: 26,304 data points,

Weather: Over 13 million data points.

Datasets like Weather provide TsMix with ample material to create a more comprehensive synthetic training set, enabling better performance without needing any original real data. The KSync synthesis method is related to our noise component in that KSync employs a novel approach to synthesize data based on white noise. KSync adds Gaussian white noise with a fixed variance to the synthetic data generated by the Gaussian Process Regressor. Similarly, in our noise synthesis process, we also incorporate Gaussian noise as one of the noise distributions.

The novelty of our method is combination of uniformly sampled Rhythm, noise, Trend components, along with providing a set of over a dozen different noise distributions for the noise and Trend. This allows us to offer any model a simulation scenario that has a probability distribution overlap with most real-world time series, while ensuring the training set is not constrained by the amount of real data available. As a result, our method performs better in scenarios with limited real data.

Based on the above, our synthetic data is particularly suitable for real-world scenarios where data is difficult to collect in large quantities or entirely, such as in cases like Illness, Exchange, ETT, or iEEG (high surgical risk). Additionally, the TsMix+KSync (Chronos) method, which uses 90% TsMix synthetic data, significantly outperforms other methods on datasets with ample data and sufficient coverage, such as the Weather dataset.

### A.9 DATASETS USED IN THE EXPERIMENT

Here, we provide all the datasets used, and to ensure balance in the number of segments across each dataset, those with over ten thousand segments were downsampled to exactly ten thousand segments. We gathered 35 publicly available time-series datasets from two prominent time-series collections, the Tslib and Monash collections Wu et al. (2023); Godahewa et al. (2021), along with various other datasets. Ultimately, this process guarantees that the baseline training set consisting of real data contains around 200,000 segments, which translates to approximately 160,000,000 time points:

- MAYO Nejedly et al. (2020)
- FNUSA Nejedly et al. (2020)
- Bern12 Andrzejak et al. (2012)
- Bern20 Martínez et al. (2020)
- Bonn Andrzejak et al. (2001)

- eeg sleep-edf
- ETT-small
- Foreign exchange_rate
- SMAP
- SMD
- weather 2020 whole year
- Individual_household_electric_power_consumption_dataset
- Traffic congestion
- australian_electricity_demand_dataset
- bitcoin_dataset_with_missing_values
- car_parts_dataset_with_missing_values
- cif_2016_dataset
- covid_deaths_dataset
- fred_md_dataset
- hospital_dataset
- kaggle_web_traffic_dataset_with_missing_values
- kaggle_web_traffic_weekly_dataset
- kdd_cup_2018_dataset_with_missing_values
- london_smart_meters_dataset_with_missing_values
- nn5_daily_dataset_with_missing_values
- nn5_weekly_dataset
- oikolab_weather_dataset
- rideshare_dataset_with_missing_values
- saugeenday_dataset
- solar_10_minutes_dataset
- solar_4_seconds_dataset
- solar_weekly_dataset
- sunspot_dataset_with_missing_values
- temperature_rain_dataset_with_missing_values
- tourism_quarterly_dataset
- tourism_yearly_dataset
- traffic_hourly_dataset
- traffic_weekly_dataset
- us_births_dataset
- vehicle_trips_dataset_with_missing_values
- weather_dataset
- wind_4_seconds_dataset
- wind_farms_minutely_dataset_with_missing_values

For the missing values, zero imputation was performed in all cases.

## A.10 SUMMARY OF NOISE DISTRIBUTIONS

A summary of each noise distribution utilized in subsubsection 3.1.2 GENERATING DIFFERENT TYPES OF NOISE:

Normal distribution: Symmetric bell-shaped dist. Params:mean mu & standard deviation sigma.

Student's t distribution: Continuous dist with heavier tails than the normal dist,characterized by degrees of freedom.

Uniform distribution:Flat dist with equal probability within a fixed interval.

Exponential distribution:Continuous dist with a sharp peak at zero and a long tail characterized by a rate parameter lambda.

Poisson distribution:Discrete dist,models the number of events occurring in a fixed interval,characterized by mean lambda.

Binomial distribution:Discrete dist ,models the number of successes in a fixed number of independent trials,characterized by the number of trials n and probability of success p.

Negative Binomial distribution:Discrete dist, models the number of failures before a fixed number of successes,characterized by the number of successes r and probability of success p.

Pareto distribution:Heavy-tailed dist, often used in economics and social sciences,characterized by scale x_m and shape alpha.

Generalized Gamma distribution: Highly flexible dist that can model a wide range of dist shapes,params: concentration c scale s and power p.

Log-Normal distribution:Models variables constrained to be positive,characterized by mean mu and standard deviation sigma of the underlying normal distribution.

Exp-LogNorm distribution:With exponential tails, using a log-normal distribution with random mean and standard deviation.

Gamma distribution: Flexible distribution often used for modeling waiting times or sums of exponentially distributed random variables characterized by shape alpha and rate beta.

Beta distribution: Continuous dist defined on the interval 0-1 often used as a prior dist in Bayesian statistics characterized by two concentration parameters alpha and beta.

Weibull distribution: Asymmetric, suitable for simulating skewed and heavy-tailed noise,characterized by scale lambda and shape k.

Rayleigh distribution: Continuous dist,models the magnitude of a two-dimensional vector,characterized by a scale parameter sigma.

## A.11 CURRENT HANDLING OF EXTREME EVENT DATA

We have observed extreme event data in the weather and energy datasets used in our study, to simulate this, we designed our synthetic data generation method to include long-tailed distributions among the types of noise distributions to simulate extreme events. Long-tailed distributions are closely associated with extreme events, as they inherently model the occurrence of rare but significant occurrences with heavier tails compared to more common distributions. All noise distributions is detailed in subsubsection 3.1.2.

The distributions that typically exhibit heavy-tailed (or long-tailed) behavior are as follows:

1) Pareto Distribution: By definition, this is a heavy-tailed distribution.

And potentially Long-tailed Distributions (The sampling function is capable of exhibiting long-tail characteristics as its input parameters vary. All the sampling parameters are initially obtained through uniform randomization, thus these distributions will exhibit long-tail noise samples with a certain probability):

2) Student's t-distribution: This distribution exhibits heavy-tail characteristics when the degrees of freedom are low.

3) Generalized Gamma Distribution: When Power p < 1, the distribution exhibits heavy-tail characteristics, and the tails become heavier as p approaches 0. When Concentration $\alpha < 1$, the distribution may also have heavier tails.

4) Gamma Distribution: When the shape parameter k is small (close to 0), the distribution exhibits heavy-tail characteristics.

Based on the above design, both our synthetic data and feature decomposer will include a certain proportion of long-tailed distributions in the noise component (with a simple estimation of occurrence probability $> 1/15$ and $< 4/15$) to simulate extreme events. The trend component will also have the same probability of exhibiting a long-tailed distribution during generation, but due to stronger smoothing, its feature in the time domain will not be as obvious. The reason behind this design is that extreme events should be defined as parts outside the rhythmic components; otherwise, if extreme events appear in the form of rhythmic data, our rhythmic generation with uniform sampling would be sufficient to cover normalized rhythmic extreme events.

Additionally, as mentioned in subsubsection 3.1.2, we performed a random y-axis inversion of our noise sampling results. This inversion can produce significant changes in noise sampling results with long-tail distributions, and it also allows the noise component of the synthetic data to simulate a wider variety of real-world scenarios. The Figure 10 shows the effect of the inversion using a Pareto distribution as an example.

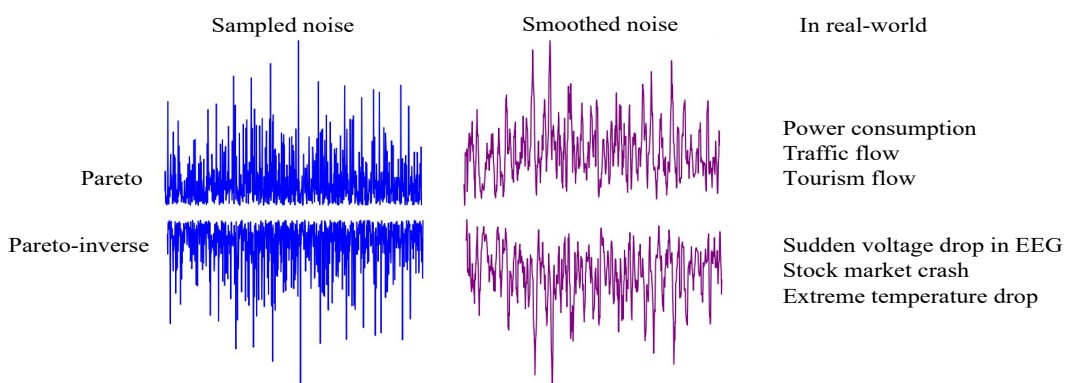

Figure 10: The effect of the noise inversion using a Pareto distribution as an example, and possible real-data scenarios that might be involved.

But still, as described in subsection A.2, since our current approach to handling extreme data involves simulating the extreme events data using a long-tailed distribution noise component, we are unable to distinguish between extreme events and noise when further decomposing the real data. Currently, we are exploring the design of additional synthesis methods to address this issue.

### A.12    OTHER POTENTIAL APPLICATIONS OF INFOBOOST'S SYNTHETIC DATA

Part from training tasks utilized in subsection 4.1 ¡Unsupervised autoencoding of Real Data Solely Trained on Synthetic Data¿ and subsection 4.2 ¡Self-supervised domain-specific prediction¿ often seen in Cross-Domain Transfer Learning, as well as the instance depicted in Figure 7 of subsection 4.3¡Case Study of Explicit Feature Extraction¿ where a feature extractor is trained to identify and separate rhythmic elements for the sake of purifying frequency domain data, there are further functions to consider:

1. Data Augmentation and Expansion: In scenarios where real-world time series datasets are limited or lack certain rare patterns, InfoBoost can generate realistic synthetic time series data, which can fill gaps and help models better understand diverse patterns, thereby improving their predictive accuracy on unseen data.

2. Privacy Protection and Compliance: For sensitive time series data (e.g., medical records, financial data), InfoBoost offers a solution by synthesizing substitute data that retains representative features while avoiding direct exposure of confidential information, complying with privacy regulations.

3. Handling Complex Noise and Trends: The InfoBoost framework excels at managing interference from multiple signal sources, noise, and capturing long-term trends. It creates synthetic data with such complexities, allowing models to become more robust and adept at parsing and forecasting actual data amidst challenging environments.

4. Unsupervised or Weakly Supervised Learning: Even in cases of scarce labeled data, InfoBoost can produce high-quality synthetic data with underlying patterns, making it possible to train models under unsupervised or weakly supervised settings to uncover significant structures and rules in time series data.

5. Zero-shot Conditional Generation: In this context, our cross-domain feature decomposer (subsection 4.3), aiming to develop a method that is not limited by the constraints of real data. The goal is to infer the probability distributions of the r_ratio, n_ratio, t_ratio of real data through data-synthesis reverse engineering. This design for predicting the missing information in real data is also presented in subsection 3.2 & Figure 7. Building upon this research, our ongoing plans focus on leveraging the capabilities of the feature decomposer to extract info: including ratios, from specific datasets and tasks without requiring fine-tuning or training on real data. This could further enable zero-shot conditional generation of specific datasets and tasks.

### A.13 EXPERIMENTS COMPUTE RESOURCES

All steps and experiments related to deep model training in this paper were conducted on a single NVIDIA GeForce 3090 GPU with 24GB memory. The batch sizes used during training were set according to the actual GPU memory consumption of each model, aiming to select the largest feasible batch size that the hardware could accommodate.

The sections in this paper that correspond to this computational resource include: ¡Universal Time Series features Extraction¿subsection 3.2, ¡Unsupervised autoencoding of Real Data Solely Trained on Synthetic Data¿subsection 4.1, ¡Self-supervised domain-specific prediction¿subsection 4.2, ¡Unsupervised autoencoding with unlimited quantity synthetic data¿subsection A.1, ¡Ablation of Rhyth, Noise and Trend components¿subsection 4.4.

Other supplementary experiments were conducted under hardware limitations, specifically on an NVIDIA RTX 4060 Ti with 12GB of VRAM.

### A.14 OPEN ACCESS TO DATA AND CODE

Due to the absence of private data and training parameters, to ensure that the acceptance of this work is not affected, we can only make all source code for InfoBoost available as an open-source library upon acceptance by any conference or journal. Currently, for experimental or verification purposes, we have anonymously released synthetic data generated using our method, ranging from lengths of 200 to 1600, in a repository: `https://anonymous.4open.science/r/InfoBoost_synth_data-5F8D/`.

