# OpenReview forum: "Are Synthetic Time-series Data Really not as Good as Real Data?"
_ICLR.cc/2025/Conference — Submitted to ICLR 2025_

### Official Review · Reviewer_x7M8 · 2024-10-26

**Soundness:** 1
**Presentation:** 1
**Contribution:** 2
**Rating:** 3
**Confidence:** 4

**Summary:**

This paper proposes a novel method for generating synthetic data that requires no information from real data yet remains highly effective for downstream tasks. By combining elements of rhythm, noise, and trend, the synthetic data is created and then applied to downstream tasks such as unsupervised learning, forecasting, and data imputation, ultimately enhancing performance in these areas.

**Strengths:**

The experimental results are intriguing, showing that synthetic data unrelated to the original dataset can effectively boost performance.

**Weaknesses:**

1.	The writing quality is quite low, with too many incomplete sentences and typos.
2.	In the fourth paragraph of the Introduction, the link between proposing a frequency-based method and addressing significant variations is weak.
3.	The author compares their method to data generation models; however, the primary purpose of data generation models is to simulate real data distribution. Since it’s impossible to replicate real data distribution without any information from actual data, this approach cannot be considered a true data generation method.
4.	Why aren’t the structures of the decomposer networks included in the main manuscript?

**Questions:**

See Weakness

---

### Official Review · Reviewer_AWR6 · 2024-11-02

**Soundness:** 1
**Presentation:** 1
**Contribution:** 1
**Rating:** 1
**Confidence:** 4

**Summary:**

The paper describes a synthetic time series data generator for
zero-shot learning of models. It is claimed not to need DL, real data
or statistics.

**Strengths:**

The goal is important: of generating simulated data to aid in learning
when data are limited. And to do that in as task-agnostic way as
possible.

The proposed method is tested in three different kinds of case
studies.

**Weaknesses:**

The writing is unclear. Concepts are not introduced before being used,
and the main learning setup and goal is not stated clearly
anywhere. Some examples in the Questions below. A couple of others: In
Fig 7 it is not clear what the fig shows and what can be concluded
from it. In Eqn 8 the notation has not been introduced.

Overstated claims: While some aspects of the proposed method may be
"non-DL" and "without the need for real data or data statistics", it
is hard to see how generalizable learning would be possible without
any data or task-relevant inductive biases whatsoever. No free lunch
applies.

Some unclarities are resolved in the supplementary material but the
main points should become clear already in the main text.

**Questions:**

1. Why is the method called non-DL?

2. How does the method guarantee generalization for new data, if it does
not use any real data or data statistics? Where do the inductive
biases come from?

3. In section 3.2: says "only needs input data". What data, and does that
not imply you need data?

4. In section 4.1 you talk about random real-world subsets. What were
they used for, precisely?

5. What is the learning task the data is generated for?

6. Example of unclearities: what does eqn 2 mean?

7. Generating trends (section 3.1.3): Seems the genrated waveforms differ
from the earlier ones only in the length of the period. How different
is the length?

8. What are the labels mentioned in Section 3.2? What broadcasting and
interpolation do you refer to?

9. Section 4.1: What is "the 'sync' group"?

10. Fig 5: too small and thin font. The table is not really understandable
based on the current explanation.

11. p. 8: What autoencoding performance do you refer to, precisely?

12. Fig 6: Is this prediction a sensible task in each domain? WHy?

---

### Official Review · Reviewer_35vG · 2024-11-03

**Soundness:** 2
**Presentation:** 3
**Contribution:** 1
**Rating:** 3
**Confidence:** 4

**Summary:**

The paper attempts to suggest that one can use exclusively. synthetic data to augment the learning of a machine learning based model. It suggests techniques to construct the time series data in a way which encompasses a wide array of signals.

**Strengths:**

It is good that they have suggested to use a wide array of signals for the synthetic data generation.

**Weaknesses:**

This is unfortunately not enough to have sufficient novelty for publication, although it can be built upon further. And it is not carefully thought out in my opinion. For example we may consider Figure 1 there is a diagrammatic separation of "Ryth", "Noise", and "Trend", in a way that these feel orthogonal constructors to an overarching to an extent, but these are not in practice. "Ryth" has elements of noise and trend added into it, "Noise" has elements of Ryth and trend added into it implicitly (if the authors don't believe so, one can simply look at a simple noise process such as random walk used to model stock prices and why that implicitly tends to drift from a non zero trend over time), and "trend" is the only the true seperator. But then why didn't you work in the frequency domain for "Rhyth" and explicitly change the low freq. domain peaks of said signal. This would be far more amenable for methods in low freq. structural engineering where you could justifiy a stronger use of this approach?

In this sense to summarise, there needs to be a proper and full mathematical justification of why "Ryth" and "Noise" are separately allowed to be modeled in addition to the infinitude of ways in which one can decompose signal construction. In addition to trend in a way of F = f + noise type of way where it is clear here "f" is a kind of low frequency trend component and "noise" is a type high frequency additional, yet pseudo-orthogonal component.

In addition to this there is not ample comparison to other time series synthesis approaches in my opinion. Since there has already been a lot of "frequency mixing" and "amplitude mixing" based time series synthesis based approaches already done in classical literature (not even considering modern literature on the topic at this point) on the topic and the authors have not demonstrated why "their" over arching approach is clearly superior. It is fine if it is also the same in my opinion, but provides a uniquely distinct window we haven't explored before but have not also seen that. If the authors could comment on why they feel their approach provides a strong and unique angle to their problem, above such traditional approaches that could be nice for a future contribution.

**Questions:**

I have no questions at the moment, but in the follow up discussion I am open to discussing responses and questions to the paper from the authors in this format.

---

### Official Review · Reviewer_unqN · 2024-11-04

**Soundness:** 1
**Presentation:** 1
**Contribution:** 1
**Rating:** 3
**Confidence:** 4

**Summary:**

The paper proposes InfoBoost, an algorithmic method for generating synthetic time series. The algorithm is based on simulating trend, periodic, and noise components and then combining them. The authors demonstrate the method's utility by adding the generated data to real data in various experimental scenarios.

**Strengths:**

- **Variety of the datasets in the experiments**. The authors evaluate their method on a wide variety of datasets.

**Weaknesses:**

- **Writing.** The paper is hard to read and needs to be better structured. For instance, it would be helpful to provide a pseudocode and a dedicated section that describes the proposed method step-by-step. Additionally, the design choices are not discussed. The paper contains many typos and requires significant polishing.
- **Methodological contribution.** The authors propose an algorithm for generating random time series signals while not discussing at all any guarantees of such an algorithm. Also, the training algorithm is not clearly presented.
- **Empirical results**. It would be great to compare the method with STS and other methods for synthetic time series generation. The empirical results, in my opinion, are not currently reproducible because the paper lacks details about it, and the source code is not available.

**Questions:**

>  data problem in DL (DL),

Typo

>  on training DL

DL model?

Many other typos in the text.

References and citations do not seem to follow ICLR guidelines.


>  non-DL approach (no need for real data to train)

“no need for real data to train” != non-DL. “no need for real data to train” means it does not require data, whereas non-DL means “does not use deep learning methods”


Is it possible to learn parameters of your simulator with simulation-based inference?

It is easy to come up with many different designs of a simulator that uses periodic-trend-noise decomposition? Why is your design better?

The empirical results seem to be very surprising in my opinion. I encourage authors to add their code and provide clear guidelines to reproduce it.

---

### Meta-Review · Area_Chair_pty1 · 2024-12-12

**Metareview:**

**(a) Summary**

This paper explores the synthetic generation of time series data, which can be a valuable contribution to time series analysis, as datasets in this domain are often insufficient. The proposed method combines three key components, Rhythm, Noise, and Trend, to generate new time series data. The effectiveness of the approach is empirically evaluated across several scenarios.


**(b) Strengths**

The paper addresses an important problem, as the generation of synthetic time series data is critical for overcoming the challenges posed by insufficient datasets in this field.

**(c) Weaknesses**

As highlighted by the reviewers, the paper has several critical weaknesses:
- **Presentation:** Numerous unclear descriptions and incorrect notations significantly deteriorate the quality and clarity of the paper.
- **Quality:** There are significant concerns regarding the design of the proposed method, including how the mixing of the three components (Rhythm, Noise, and Trend) is validated, the theoretical properties of the method, and its training process.
- **Empirical evaluation:** The experimental analysis is insufficient. For instance, comparisons with traditional approaches are absent, making it difficult to convincingly demonstrate the superiority of the proposed method.

**(d) Reason of the decision**

The weaknesses identified above are critical limitations and, individually, are sufficient for rejection.

**Additional Comments On Reviewer Discussion:**

No rebuttal was provided, and all reviewers recommended rejecting the paper without offering any further specific comments.

---

### Decision · Program_Chairs · 2025-01-22

Reject